# UNSUPERVISED LEARNING OF NEUROSYMBOLIC ENCODERS

## ABSTRACT

We present a framework for the unsupervised learning of neurosymbolic encoders, i.e., encoders obtained by composing neural networks with symbolic programs from a domain-specific language. Such a framework can naturally incorporate symbolic expert knowledge into the learning process and lead to more interpretable and factorized latent representations than fully neural encoders. Also, models learned this way can have downstream impact, as many analysis workflows can benefit from having clean programmatic descriptions. We ground our learning algorithm in the variational autoencoding (VAE) framework, where we aim to learn a neurosymbolic encoder in conjunction with a standard decoder. Our algorithm integrates standard VAE-style training with modern program synthesis techniques. We evaluate our method on learning latent representations for real-world trajectory data from animal biology and sports analytics. We show that our approach offers significantly better separation than standard VAEs and leads to practical gains on downstream tasks.

## 1 INTRODUCTION

Advances in unsupervised learning have enabled the discovery of latent structures in data from a variety of domains, such as image data (Dupont, 2018), sound recordings (Calhoun et al., 2019), and tracking data (Luxem et al., 2020). For instance, a common approach is to use encoder-decoder frameworks, such as variational autoencoders (VAE) (Kingma & Welling, 2014), to identify a low-dimensional latent representation from the raw data that could contain disentangled factors of variation (Dupont, 2018) or semantically meaningful clusters (Luxem et al., 2020). Such approaches typically employ complex mappings based on neural networks, which can make it difficult to explain how the model assigns inputs to latent representations (Zhang et al., 2020).

To address this issue, we introduce *unsupervised neurosymbolic representation learning*, where the goal is to find a programmatically interpretable representation (as part of a larger neurosymbolic representation) of the raw data. We consider programs to be differentiable, symbolic models instantiated using a domain-specific language (DSL), and use neurosymbolic to refer to blendings of neural and symbolic. Neurosymbolic encoders can offer a few key benefits. First, since the DSL reflects structured domain knowledge, they can often be human-interpretable (Verma et al., 2018; Shah et al., 2020). Second, by leveraging the inductive bias of the DSL, neurosymbolic encoders can potentially offer more factorized or well-separated representations of the raw data (i.e., the representations are more semantically meaningful), which has been observed in studies that used hand-crafted programmatic encoders (Zhan et al., 2020).

Our learning algorithm is grounded in the VAE framework (Kingma & Welling, 2014; Mnih & Gregor, 2014), where the goal is to learn a neurosymbolic encoder coupled with a standard neural decoder. A key challenge is that the space of programs is combinatorial, which we tackle via a tight integration between standard VAE training with modern program synthesis methods (Shah et al., 2020). We further show how to incorporate ideas from adversarial information factorization (Creswell et al., 2017) and enforcing capacity constraints (Burgess et al., 2017; Dupont, 2018) in order to mitigate issues such as posterior and index collapse in the learned representation.

We evaluate our neurosymbolic encoding approach on multiple behavior analysis domains, where the data are from challenging real-world settings and cluster interpretability is important for domain experts. Our contributions are:

- We propose a novel unsupervised approach to train neurosymbolic encoders, to result in a programmatically interpretable representation of data (as part of a neurosymbolic representation).

- We show that our approach can significantly outperform purely neural encoders in extracting semantically meaningful representations of behavior, as measured by standard unsupervised metrics.

- We further explore the flexibility of our approach, by showing that performance can be robust across different DSL designs by domain experts.

- We showcase the practicality of our approach on downstream tasks, by incorporating our approach into a state-of-the-art self-supervised learning approach for behavior analysis (Sun et al., 2021b).

## 2 BACKGROUND

### 2.1 VARIATIONAL AUTOENCODERS

We build on VAEs (Kingma & Welling, 2014; Mnih & Gregor, 2014), a latent variable modeling framework shown to learn effective latent representations (also called encodings/embeddings) (Higgins et al., 2016; Zhao et al., 2017; Yingzhen & Mandt, 2018) and can capture the generative process (Oord et al., 2017; Vahdat & Kautz, 2020; Zhan et al., 2020). VAEs introduce a latent variable $\mathbf{z}$, an encoder $q_\phi$, a decoder $p_\theta$, and a prior distribution $p$ on $\mathbf{z}$. $\phi$ and $\theta$ are the parameters of the $q$ and $p$ respectively, often instantiated with neural networks. The learning objective is to maximize the evidence lower bound (ELBO) of the data log-likelihood:

$$\text{ELBO} := \mathbb{E}_{q_\phi(\mathbf{z}|\mathbf{x})}\big[\log p_\theta(\mathbf{x}|\mathbf{z})\big] - D_{KL}\big(q_\phi(\mathbf{z}|\mathbf{x})||p(\mathbf{z})\big) \leq \log p(\mathbf{x}). \quad (1)$$

The first term in Eq. 1 is the log-density assigned to the data, while the second term is the KL-divergence between the prior and approximate posterior of $\mathbf{z}$. Latent representations $\mathbf{z}$ are often continuous and modeled with a Gaussian prior, but $\mathbf{z}$ can be modeled to contain discrete dimensions as well (Kingma et al., 2014; Hu et al., 2017; Dupont, 2018). Our experiments are focused on behavioral tracking data in the form of trajectories, and so in practice we utilize a trajectory variant of VAEs (Co-Reyes et al., 2018; Zhan et al., 2020; Sun et al., 2021b), described in Section 3.4.

One challenge with VAEs (and deep encoder-decoder models in general) is that while the model is expressive, it is often difficult to interpret what is encoded in the latent representation $\mathbf{z}$. Common approaches include taking traversals in the latent space and visualizing the resulting generations (Burgess et al., 2017), or post-processing the latent variables using techniques such as clustering (Luxem et al., 2020). Such techniques are post-hoc and thus cannot guide (in an interpretable way) the encoder to be biased towards a family of structures. Some recent work have studied how to impose structure in the form of graphical models or dynamics in the latent space (Johnson et al., 2016; Deng et al., 2017), and our work can be thought of as a first step towards imposing structure in the form of symbolic knowledge encoded in a domain specific programming language.

### 2.2 SYNTHESIS OF DIFFERENTIABLE PROGRAMS

Our approach utilizes recent work on the synthesis of differentiable programs (Shah et al., 2020; Valkov et al., 2018), where one learns both the discrete structure of the symbolic program (analogous to the architecture of a neural network) as well as differentiable parameters within that structure. Our formulation of this problem closely follows that of Shah et al. (2020). We use a domain-specific functional programming language (DSL), generated with a context-free grammar (see Figure 2 for an example). Programs are represented as a pair $(\alpha, \psi)$, where $\alpha$ is a discrete program architecture and $\psi$ are its real-valued parameters. We denote $\mathcal{P}$ as the space of symbolic programs (i.e. programs with complete architectures). The semantics of a program $(\alpha, \psi)$ are given by a function $[\![\alpha]\!](x, \psi)$, which is guaranteed by the semantics of the DSL to be differentiable in both $x$ and $\psi$.

Like Shah et al. (2020), we pose the problem of learning differentiable programs as search through a directed program graph $\mathcal{G}$. The graph $\mathcal{G}$ models the top-down construction of program architectures $\alpha$ through the repeated firing of rules of the DSL grammar, starting with an *empty* architecture (represented by the "start" nonterminal of the grammar). The *leaf nodes* of $\mathcal{G}$ represent programs with *complete* architectures (no nonterminals). Thus, $\mathcal{P}$ is the set of programs in the leaf nodes of $\mathcal{G}$. The other nodes in $\mathcal{G}$ contain programs with *partial* architectures (has at least one nonterminal). We interpret a program in a non-leaf node as being neurosymbolic, by viewing its nonterminals as

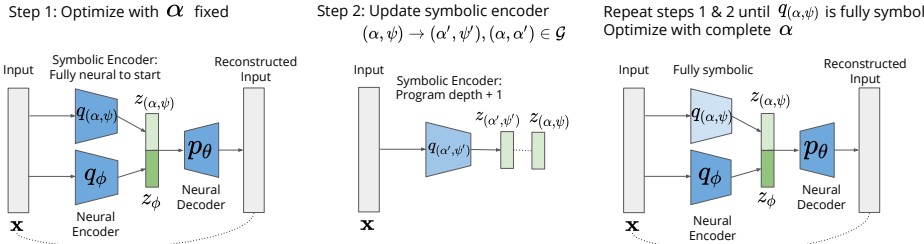

Figure 1: Sketch of Algorithm 1 (Section 3.1). The symbolic encoder is initially fully neural. We alternate between VAE training with the program architecture fixed (Step 1), and supervised program learning to increase the depth of the program by 1 (Step 2). Once we reach a symbolic program, we train the model one last time to learn all the parameters. The color (in terms of lightness) of the symbolic encoder corresponds to the encoder becoming more symbolic over time.

representing neural networks with free parameters. The root node in $\mathcal{G}$ is the empty architecture $\alpha_0$, interpreted as a fully neural program. An edge $(\alpha, \alpha')$ exists in $\mathcal{G}$ if one can obtain $\alpha'$ from $\alpha$ by applying a rule in the DSL that replaces a nonterminal in $\alpha$.

Program synthesis in this problem setting equates to searching through $\mathcal{G}$ to find the optimal complete program architecture, and then learning corresponding parameters $\psi$, i.e., to find the optimal $(\alpha, \psi)$ that minimizes a combination of standard training loss (e.g., classification error) and structural loss (preferring "simpler" $\alpha$'s). Shah et al. (2020) evaluate multiple strategies for solving this problem and finds *informed search using admissible neural heuristics* to be the most efficient strategy (see appendix). Consequently, we adopt this algorithm for our program synthesis task.

## 3 NEUROSYMBOLIC ENCODERS

The structure of our neurosymbolic encoder is shown in the right diagram of Figure 1. The latent representation $\mathbf{z} = [\mathbf{z}_\phi, \mathbf{z}_{(\alpha, \psi)}]$ is partitioned into neurally encoded $\mathbf{z}_\phi$ and programmatically encoded $\mathbf{z}_{(\alpha, \psi)}$. This approach boasts several advantages:

- The symbolic component of the latent representation is programmatically interpretable.
- The neural component can encode any residual information not captured by the program, which maintains the model's capacity compared to standard deep encoders.
- By incorporating a modular design, we can leverage state-of-the-art learning algorithms for both differentiable encoder-decoder training and program synthesis.

We denote $q_\phi$ and $q_{(\alpha, \psi)}$ as the neural and symbolic encoders respectively (see Figure 1), where $\mathbf{z}_\phi \sim q_\phi(\cdot|\mathbf{x})$ and $\mathbf{z}_{(\alpha, \psi)} \sim q_{(\alpha, \psi)}(\cdot|\mathbf{x})$. $q_\phi$ is instantiated with a neural network, but $q_{(\alpha, \psi)}$ is a differentiable program with architecture $\alpha$ and parameters $\psi$ in some program space $\mathcal{P}$ defined by a DSL. Given an unlabeled training set of $\mathbf{x}$'s, the VAE learning objective in Eq. 1 then becomes:

$$
\max_{\phi, (\alpha, \psi), \theta} \quad \mathbb{E}_{q_\phi(\mathbf{z}_\phi|\mathbf{x})q_{(\alpha, \psi)}(\mathbf{z}_{(\alpha, \psi)}|\mathbf{x})} \Big[ \underbrace{\log p_\theta(\mathbf{x}|\mathbf{z}_\phi, \mathbf{z}_{(\alpha, \psi)})}_{\text{reconstruction loss}} \Big]
$$
$$
- \underbrace{D_{KL}\big(q_\phi(\mathbf{z}_\phi|\mathbf{x})||p(\mathbf{z}_\phi)\big)}_{\text{regularization for neural latent}} - \underbrace{D_{KL}\big(q_{(\alpha, \psi)}(\mathbf{z}_{(\alpha, \psi)}|\mathbf{x})||p(\mathbf{z}_{(\alpha, \psi)})\big)}_{\text{regularization for symbolic latent}}. \tag{2}
$$

Compared to the standard VAE objective in Eq. 1 for a single neural encoder, Eq. 2 has separate KL-divergence terms for the neural and programmatic encoders.

### 3.1 LEARNING ALGORITHM

The challenge with solving for Eq. 2 is that while $(\phi, \psi, \theta)$ can be optimized via back-propagation with $\alpha$ fixed, optimizing for $\alpha$ is a discrete optimization problem. Since it is difficult to jointly optimize over both continuous and discrete spaces, we take an iterative, alternating optimization approach. We start with a fully neural program (one with empty architecture $\alpha_0$) trained using standard differentiable optimization (Figure 1, Step 1). We then gradually make it more symbolic (Figure 1, Step 2) by finding a program that is a child of the current program in $\mathcal{G}$ (more symbolic

by construction of $\mathcal{G}$) that outputs as similar to the current latent representations as possible:

$$\min_{\alpha':(\alpha,\alpha')\in\mathcal{G},\ \psi'} \mathcal{L}_{\text{supervised}}\big(q_{(\alpha,\psi)}(\mathbf{x}), q_{(\alpha',\psi')}(\mathbf{x})\big), \tag{3}$$

which can be viewed as a form of distillation (from less symbolic to more symbolic programs) via matching the input/output behavior. We solve Eq. 3 by enumerating over all child programs and selecting the best one, which is similar to iteratively-deepened depth-first search in Shah et al. (2020) (see appendix). We alternate between optimizing Eq. 2 and Eq. 3 until we obtain a complete program. Algorithm 1 outlines this procedure and is guaranteed to terminate if $\mathcal{G}$ is finite by specifying a maximum program depth.

We chose this optimization procedure for two reasons. First, it maximally leverages state-of-the-art tools in both differentiable latent variable modeling (VAE-style training) and supervised program synthesis, leading to tractable algorithm design. Second, this procedure never makes a drastic change to the program architecture, leading to relatively stable learning behavior across iterations.

---

**Algorithm 1** Learning a neurosymbolic encoder

1: **Input**: program space $\mathcal{P}$, program graph $\mathcal{G}$
2: initialize $\phi, \psi, \theta, \alpha = \alpha_0$ (empty architecture)
3: **while** $\alpha$ is not complete **do**
4:    $\phi, \psi, \theta \leftarrow$ optimize Eq. 2 with $\alpha$ fixed
5:    $(\alpha, \psi) \leftarrow$ optimize Eq. 3
6: **end while**
7: $\phi, \psi, \theta \leftarrow$ optimize Eq. 2 with complete $\alpha$
8: **Return**: encoder $\{q_\phi, q_{(\alpha,\psi)}\}$

**Algorithm 2** Learning a neurosymbolic encoder with $k$ programs

1: **Input**: program space $\mathcal{P}$, program graph $\mathcal{G}$, $k$
2: **for** $i = 1..k$ **do**
3:    fix programs $\{q_{(\alpha_1,\psi_1)}, \ldots, q_{(\alpha_{i-1},\psi_{i-1})}\}$
4:    execute Algorithm 1 to learn $q_{(\alpha_i,\psi_i)}$
5:    remove $q_{(\alpha_i,\psi_i)}$ from $\mathcal{P}$ to avoid redundancies
6: **end for**
7: **Return**: encoder $\{q_\phi, q_{(\alpha_1,\psi_1)}, \ldots, q_{(\alpha_k,\psi_k)}\}$

---

### 3.2 LEARNING MULTIPLE PROGRAMS

The interpretability of latent representations induced by symbolic encoders $q_{(\alpha,\psi)}$ ultimately depends on the DSL. For instance, a program that encodes to one of ten classes may not be very interpretable if it involves a matrix multiplication within the program. Instead, we learn *binary* programs that encode sequences into one of two classes (using binary cross-entropy for $\mathcal{L}_{\text{supervised}}$, a uniform prior on $\mathbf{z}_{(\alpha,\psi)}$, and Gumbel-Softmax (Jang et al., 2017) to sample from the posterior). Figures 4a & 4b depict learned binary programs that encode mice trajectories and their interpretations.

To encode more than two classes, we can simply learn multiple binary programs by extending Eq. 2 to sum over $\mathcal{L}_{\text{supervised}}$ for $k$ symbolic programs $\{q_{(\alpha_1,\psi_1)}, \ldots, q_{(\alpha_k,\psi_k)}\}$ and corresponding latent representations $\{\mathbf{z}_{(\alpha_1,\psi_1)}, \ldots, \mathbf{z}_{(\alpha_k,\psi_k)}\}$. This results in $2^k$ classes and a solution space that now scales exponentially (e.g. $|\mathcal{P}|^k$ if using exhaustive enumeration). Algorithm 2 outlines our greedy solution that reuses Algorithm 1 by iteratively learning one symbolic program at a time. We leave the exploration of more sophisticated search methods as future work.

### 3.3 DEALING WITH POSTERIOR AND INDEX COLLAPSE

Deep latent variable models, especially those with discrete latent variables, are notoriously prone to both posterior (Bowman et al., 2015; Chen et al., 2016b; Oord et al., 2017) and index (Kaiser et al., 2018) collapse. Since our algorithms optimize for such models repeatedly, we can be more susceptible to these failure modes. There are many approaches available for tackling both these issues, but we emphasize that these contributions are orthogonal to ours; as techniques for preventing posterior and index collapse improve, so will the robustness of our algorithm. Below, we summarize two strategies that we found to work well in our setting.

**Adversarial information factorization** Creswell et al. (2017) introduces an adversarial network $A_\omega$ that aims to predict $\mathbf{z}_{(\alpha,\psi)}$ from $\mathbf{z}_\phi$. Maximizing this adversarial loss can prevent index collapse, where all data is encoded into the same class, as doing so would would fail to fool the adversary.

$$\max_{\phi,(\alpha,\psi),\theta} \mathbb{E}_{q_\phi(\mathbf{z}_\phi|\mathbf{x})q_{(\alpha,\psi)}(\mathbf{z}_{(\alpha,\psi)}|\mathbf{x})} \Big[ \log p_\theta(\mathbf{x}|\mathbf{z}_\phi, \mathbf{z}_{(\alpha,\psi)}) + \underbrace{\min_\omega \mathcal{L}_{\text{adv}}\big(A_\omega(\mathbf{z}_\phi), \mathbf{z}_{(\alpha,\psi)}\big)}_{\text{adversary}} \Big]$$
$$- D_{KL}\big(q_\phi(\mathbf{z}_\phi|\mathbf{x})||p(\mathbf{z}_\phi)\big) - D_{KL}\big(q_{(\alpha,\psi)}(\mathbf{z}_{(\alpha,\psi)}|\mathbf{x})||p(\mathbf{z}_{(\alpha,\psi)})\big) \tag{4}$$

**Channel capacity constraint** (Burgess et al., 2017; Dupont, 2018) forces the KL-divergence terms to match capacities $C_\phi$ and $C_{(\alpha,\psi)}$. Since the KL-divergence is an upper bound on the mutual information between latent variables and the data (Kim & Mnih, 2018; Dupont, 2018), this encourages the latent variables to encode information and aims to prevent posterior collapse.

$$\max_{\phi,(\alpha,\psi),\theta} \mathbb{E}_{q_\phi(\mathbf{z}_\phi|\mathbf{x})q_{(\alpha,\psi)}(\mathbf{z}_{(\alpha,\psi)}|\mathbf{x})} \Big[ \log p_\theta(\mathbf{x}|\mathbf{z}_\phi,\mathbf{z}_{(\alpha,\psi)}) \Big] - \gamma_\phi |D_{KL}\big(q_\phi(\mathbf{z}_\phi|\mathbf{x})||p(\mathbf{z}_\phi)\big) - C_\phi|$$
$$- \gamma_{(\alpha,\psi)} |D_{KL}\big(q_{(\alpha,\psi)}(\mathbf{z}_{(\alpha,\psi)}|\mathbf{x})||p(\mathbf{z}_{(\alpha,\psi)})\big) - C_{(\alpha,\psi)}| \tag{5}$$

In our algorithms, we augment our initial objective in Eq. 2 with Eq. 4 and Eq. 5.

## 3.4 Instantiation for Sequential Domains

The objective in Eq. 2 describes a general problem that is applicable to any domain. In our experiments, we focus on the sequential domain of trajectory data. Trajectory data is often used in scientific applications where interpretability is desirable, such as behavior discovery (Luxem et al., 2020; Hsu & Yttri, 2020). The ability to easily explain the learned latent representation using programs can help domain experts better understand the structure in their data. Additionally, trajectory data is often low dimensional for each timestamp, which helps experts encode domain knowledge into the DSL more easily (Shah et al., 2020; Sun et al., 2021b; Zhan et al., 2020).

In this domain, $\mathbf{x}$ is a trajectory of length $T$: $\mathbf{x} = \{x_1, \ldots, x_T\}$. We then factorize the log-density in Eq. 2 as a product of conditional probabilities:

$$\log p_\theta(\mathbf{x}|\mathbf{z}_\phi,\mathbf{z}_{(\alpha,\psi)}) = \sum_{t=1}^{T} \log p_\theta(x_t|x_{<t},\mathbf{z}_\phi,\mathbf{z}_{(\alpha,\psi)}). \tag{6}$$

When $q_\phi$ and $p_\theta$ are instantiated with recurrent neural networks (RNN), the model is more commonly known as a trajectory-VAE (TVAE) Co-Reyes et al. (2018); Zhan et al. (2020); Sun et al. (2021b).

As symbolic encoder $q_{(\alpha,\psi)}$ maps sequences to vectors, we adopt a DSL (Figure 2) previously used for sequence classification (Shah et al., 2020). Our DSL is purely functional and contains both basic algebraic operations and parameterized library functions. Domain experts can easily augment the DSL with their own functions, such as selection functions that select subsets of features that they deem potentially important. We ensure that all programs in our DSL are differentiable, utilizing a smooth approximation of the nondifferentiable **if-then-else** construct (Shah et al., 2020). Figures 4a & 4b depict example programs in our DSL (full details in the appendix).

$$\alpha \quad ::= \quad x \mid \oplus(\alpha_1, \ldots, \alpha_k) \mid \oplus_\theta(\alpha_1, \ldots, \alpha_k)$$
$$\textbf{if } \alpha_1 \textbf{ then } \alpha_2 \textbf{ else } \alpha_3 \mid \textbf{sel}_S \, x \mid \textbf{mapaverage } (\textbf{fun } x_1.\alpha_1) \, x$$

Figure 2: Our DSL for sequential domains, similar to the one used in Shah et al. (2020). $x$, $\oplus$, and $\oplus_\theta$ represent inputs, basic algebraic operations, and parameterized library functions, respectively. **fun** $x.e(x)$ represents a function that evaluates an expression $e(x)$ over the input $x$. **sel**$_S$ selects a subset $S$ of the dimensions of the input $x$. **mapaverage** $g \, x$ applies the function $g$ to every element of the sequence $x$ and returns the average of the results. We employ a different approximation of the **if-then-else** construct.

## 4 Experiments

We study our proposed approach on sequential trajectory data from a synthetic dataset to first provide intuition for our algorithm, and then on real-world datasets in neuroscience and sports analytics.

## 4.1 Experiments with Synthetic Dataset

We generate a synthetic dataset of trajectories and run our algorithm to demonstrate the following:

- Programs can capture factors of variation in the data (in our case, 2 discrete factors).
- Information pertaining to captured factors of variation are extracted out of the latent space.

We generate synthetic trajectories by sampling initial positions and velocities from a Gaussian distribution and introducing 2 ground-truth factors of variation as large external forces in the posi-

tive/negative x/y directions that affect velocity, totalling to 4 discrete classes. Velocities are fixed for the entire trajectory, but we also sample small Gaussian noise at each timestep. We generate 10k/2k/2k trajectories of length 25 for train/validation/test. Figure 3a shows 50 trajectories from the training set. Full details of the synthetic dataset are included in the appendix.

We visualize the neural latent space in 2 dimensions of a TVAE with 0, 1, and 2 learned programs in Figure 3bcd. The initial TVAE latent space contains 4 clusters corresponding to the 4 ground-truth classes in Figure 3c. After our algorithm learns the first program that thresholds the final x-position, the resulting latent space in Figure 3d captures the other factor of variation as 2 clusters corresponding to the final y-positions. Lastly, when our algorithm learns a second program that thresholds the final y-position, the resulting latent space in Figure 3e no longer contains any clear clustering, as we've successfully extracted the 4 ground-truth classes with our 2 programs.

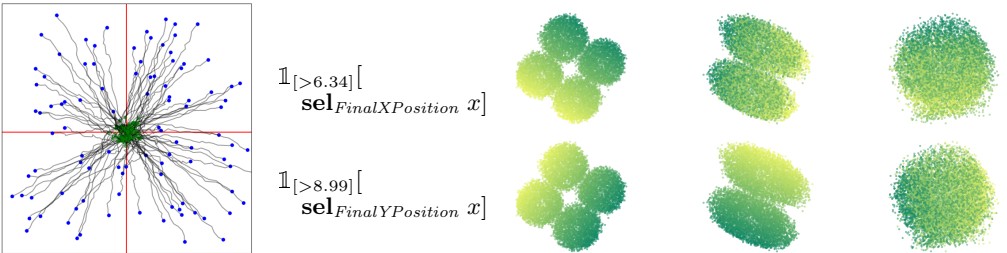

(a) 50 synthetic trajectories  (b) learned programs  (c) $\mathbf{z}_\phi$, 0 programs  (d) $\mathbf{z}_\phi$, 1 program  (e) $\mathbf{z}_\phi$, 2 programs

Figure 3: **(a)** Trajectories in synthetic training set. Initial/final positions are indicated in green/blue. Red lines delineate ground-truth classes, based on final positions. **(b)** $k = 2$ learned binary programs using our algorithm. The first program (top) thresholds the final x-position while the second program (bottom) thresholds the final y-position. **(c, d, e)** Neural latent variables reduced to 2 dimensions. Top/bottom rows are colored by final x/y-positions respectively (green/yellow is positive/negative). **(c)** Clusters in TVAE neural latent space correspond to 4 ground-truth classes. **(d)** After learning the first program, the neural latent space contains clusters only based on the final y-position. **(e)** After learning the second program, all 4 ground-truth classes have been extracted as programs and the remaining neural latent space contains no clear clustering.

## 4.2 EXPERIMENTS WITH REAL-WORLD DATASETS

### 4.2.1 DATASETS

**CalMS21.** Our primary real-world dataset is the CalMS21 dataset (Sun et al., 2021a), containing trajectories of socially interacting mice captured for neuroscience experiments. Each frame contains 7 tracked keypoints for each of two mice. The dataset has one set of unlabeled tracking data, which we use to train our neurosymbolic encoder, and another set annotated for 4 behaviors, which we use to evaluate our programs. Specifically, our evaluation uses labels from the test split of the CalMS21 classification task. We have 231k/52k/262k trajectories of length 21 for train/val/test. The features in our DSL are selected by a domain expert based on the attributes from (Segalin et al., 2020).

**Basketball.** We use the same basketball dataset as in (Shah et al., 2020; Zhan et al., 2020) that tracks professional basketball players. Each trajectory is of length 25 over 8 seconds and contains the $xy$-positions of 10 players. We split trajectories into two by grouping offensive and defensive players (5 each), effectively doubling the dataset size. We evaluate our algorithm and the baselines with respect to the labels of offensive/defensive players. Our DSL includes additional domain features like player speed and distance-to-basket. In total, we have 177k/31k/27k trajectories for train/val/test.

### 4.2.2 QUANTITATIVE EVALUATION SETUP

**Baselines.** We compare our model containing a neurosymbolic encoder against other approaches based on VAEs and its variations. In particular, we compare against VAE, VAE with K-means loss used in (Ma et al., 2019; Luxem et al., 2020), and Beta-VAE (Burgess et al., 2017). These models have a fully neural encoder and learn continuous latent representations, which we can then use to produce clusters with K-means clustering (Lloyd, 1982). Additionally, we compare against models which produce discrete latent clusterings, such as JointVAEs (Dupont, 2018) and VQ-VAEs (Oord et al., 2017). We use the TVAE version of all baselines (details included in the appendix).

$$\mathbb{1}_{[>-7.02]} \begin{bmatrix} \textbf{mapaverage (fun } x_t. \\ \quad \textbf{multiply } (ResidentSpeedAffine_{[-6.28];-8.28}(x_t), \\ \quad NoseTailDistAffine_{[.042];-9.06}(x_t)) \; x \end{bmatrix}$$

(a) Program learned using CalMS21 DSL 1, resulting NMI 0.428. Since speed is positive, the first term is always negative. One cluster thus generally consists of trajectories where the mice are further apart, such that the second term is positive, and the negative product is less than the threshold. The other cluster generally occurs when the mice are close together, the second term is negative, and the product will be positive.

$$\mathbb{1}_{[>-5.68]} \begin{bmatrix} \textbf{mapaverage (fun } x_t. \\ \quad \textbf{add } (ResidentAxisRatioAffine_{[-7.95];-7.14}(x_t), \\ \quad BoundingBoxIOUAffine_{[-16.55];5.87}(x_t)) \; x \end{bmatrix}$$

(b) Program learned using CalMS21 DSL 2, resulting NMI 0.320. The axis ratio is the ratio of major axis length and minor axis length of an ellipse fitted to the mouse keypoints. The second term measures the bounding box overlap between mice, and is zero when the mice are far apart. It follows that one cluster generally contains trajectories when the mice has larger bounding box overlaps or if the resident axis ratio is large. The other cluster thus contains trajectories where the mice bounding boxes do not overlap, and resident body is compact.

Figure 4: Learned programs on CalMS21. The subscripts represents the learned weights and biases, in particular, the brackets contain the weights for the affine transformation followed by the bias.

| Model | CalMS21 | | | Basketball | | |
|---|---|---|---|---|---|---|
| | Purity | NMI | RI | Purity | NMI | RI |
| Random assignment | .597 | .000 | .536 | .500 | .000 | .500 |
| TVAE | .598 | .089 | .564 | .501 | .001 | .500 |
| TVAE+KMeans loss | .605 | .118 | .573 | .501 | .001 | .500 |
| JointVAE | .597 | .019 | .537 | .560 | .034 | .507 |
| VQ-TVAE | .601 | .124 | .588 | .572 | .016 | .511 |
| Beta-TVAE | .616 | .115 | .589 | .565 | .013 | .509 |
| Ours (1 program) | .706 | .423 | .694 | .596 | .027 | .518 |
| Ours (2 programs) | .725 | .320 | .648 | .561 | .033 | .507 |
| Ours (3 programs) | .756 | .314 | .633 | .584 | .022 | .514 |

Table 1: Median purity, NMI, and RI on CalMS21 and Basketball compared to human-annotated labels (3 runs). Experiment hyperparameters are included in the appendix.

**Metrics.** Unlike in the synthetic setting, we do not have ground truth programs in the real-world datasets. We thus evaluate our programs quantitatively using standard cluster metrics relative to human-defined labels. In particular, we use Purity (Schütze et al., 2008), Normalized Mutual Information (NMI) (Zhang et al., 2006), and Rand Index (RI) (Rand, 1971). Purity measures the extent to which clusters contain a single human-defined class. NMI scales with the mutual information between two cluster assignments. RI measures similarity between our clusters and human labels. We report the median of three runs, and include a random baseline that assigns a class randomly to each sequence. More details, including the standard deviation and the ELBO, are in the appendix.

### 4.2.3 RESEARCH QUESTIONS

**Are the clusters created with our programs meaningful?** We compare clusters produced by our neurosymbolic encoder with fully neural autoencoding baselines (Table 1), measured against human-annotated behaviors. For CalMS21, we observe that our method consistently outperforms the baselines in all three cluster metrics. Our method is able to do this by leveraging a DSL which encodes domain knowledge such as behavior attributes that are useful for identifying behaviors. We note that the purity increases as the number of programs (thus clusters) increase, while NMI and RI decreases. This implies our method with two clusters best correspond to CalMS21 behaviors, but the other clusters found by our method may still be useful for domain experts. For Basketball, our method improves slightly with respect to purity, but is overall comparable with the baselines.

We further study the programs and clusters produced by our algorithm for the CalMS21 dataset, through a qualitative study with a domain expert in behavioral neuroscience. In the single program case, the domain expert classified the discovered clusters as when the mice are interacting and when there are no interaction. They noted that this is based on distance between the mice, which is consistent with our program (Figure 4a) using distance between nose of resident and tail of intruder. For two programs, there are a total of four clusters, with two clusters each corresponding to no in-

| Model | CalMS21 (DSL 1) | | | CalMS21 (DSL 2) | | | CalMS21 (DSL 3) | | |
|---|---|---|---|---|---|---|---|---|---|
| | Purity | NMI | RI | Purity | NMI | RI | Purity | NMI | RI |
| Ours (1 program) | .706 | .423 | .694 | .689 | .364 | .681 | .649 | .325 | .616 |
| Ours (2 programs) | .725 | .320 | .648 | .715 | .359 | .673 | .664 | .324 | .634 |

Table 2: Median purity, NMI, and RI on CalMS21 of our algorithms with DSLs selected by three domain experts compared to human-annotated labels (3 runs). DSL1 corresponds to Table 1.

| Model | CalMS21 | | | Basketball | | |
|---|---|---|---|---|---|---|
| | Purity | NMI | RI | Purity | NMI | RI |
| TVAE | .598 | .089 | .564 | .501 | .001 | .500 |
| TVAE (w/ features) | .597 | .103 | .570 | .565 | .012 | .508 |
| VQ-TVAE | .601 | .124 | .588 | .571 | .016 | .511 |
| VQ-TVAE (w/ features) | .608 | .114 | .601 | .525 | .002 | .501 |
| Beta-TVAE | .616 | .115 | .589 | .566 | .013 | .509 |
| Beta-TVAE (w/ features) | .612 | .096 | .571 | .563 | .011 | .508 |

Table 3: Median purity, NMI, and RI on CalMS21 and Basketball compared to human-annotated labels (3 runs) for baseline with trajectory inputs only, and baseline with trajectory features added.

teraction and interaction. For the interaction clusters, the domain expert was further able to identify sniff tail behavior as one of the clusters. In this case, the programs found were based on intruder head body angle, resident nose and intruder tail distance, and resident nose and intruder nose distance. The domain expert found the three program case to be more difficult to interpret, but was able to identify clusters corresponding to sniff tail, resident exploration, interaction facing the same direction (ex: mounting), and interaction facing opposite directions (ex: face-to-face sniffing).

**How sensitive is our approach to different DSL choices?** To study the effect of domain expert variation on our approach, we worked with different domain experts to construct alternate DSLs for studying mouse social behavior on CalMS21. While there is some variation in median cluster metrics, our approach consistently outperforms other baseline approaches that contain fully neural encoders for all three DSLs (Table 2). Comparing some learned programs from two DSLs (Figures 4a, 4b), both contain a term that is correlated with whether the mice is interacting (distance and bounding box overlap), and another term on resident speed (mouse tends to be more stretched when they are moving quickly). A full list of features selected by domain experts is in the appendix.

**What happens if we simply encode DSL features as part of input trajectories?** Since our method uses features from domain experts as part of our DSL, we additionally experiment with providing the same features as input to the baseline models during training (Table 3). For both CalMS21 and Basketball, providing the features as additional inputs to baseline has comparable performance to using input trajectory data alone. In contrast, by using the features more explicitly as part of the DSL in our neurosymbolic encoders, we are able to produce clusters with a better separation between behavior classes based on cluster metrics (as was seen in Table 1).

**Are the programs useful for downstream tasks?** We integrate the learned programs from our neurosymbolic encoder into the task programming framework (Sun et al., 2021b), a state-of-the-art self- and programmatically-supervised learning approach. This framework uses expert provided programs to train a trajectory representation, which can be applied to behavior analysis. Here, rather than using hand-crafted programs, we instead use the learned programs from our approach, so that experts would only need to provide the DSL. We evaluate on the same mouse dataset (Segalin et al., 2020) as task programming. Using only one program found using our approach, we are able to achieve comparable performance to 10 expert-written programs on the behavior classification task

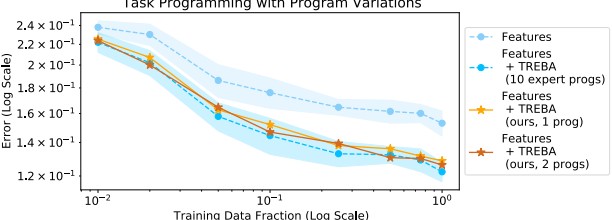

Figure 5: Applying symbolic encoders for self-supervision. "Features" is baseline w/o self-supervision. "TREBA" is a self-supervised approach, using either expert-crafted programs or our symbolic encoders as the weak-supervision rules. The shaded region is std dev over 9 repeats. The std dev for our approach (not shown) is comparable.

studied in Sun et al. (2021b) (Figure 5). This demonstrates that programs found by our approach can be applied effectively to downstream behavior analysis tasks such as task programming.

## 5 OTHER RELATED WORK

**Interpretable latent variable models.** Latent representations, especially those that are human-interpretable, can help us understand the structure of data. These models may learn disentangled factors (Higgins et al., 2016; Chen et al., 2016a) or semantically meaningful clusters (Ma et al., 2019) using unsupervised learning approaches. These approaches are often grounded in the VAE framework (Kingma & Welling, 2014). Similar to our programs which produce a discrete code, there are other VAE variations which also learns a discrete latent representation, such as Joint-VAE (Dupont, 2018), Discrete VAE (Rolfe, 2016), and VQ-VAE (Oord et al., 2017). In particular, JointVAE models also learns a discrete and continuous representation. However, these approaches use fully neural encoders. To the best of our knowledge, our work is the first to propose neurosymbolic encoders, where the symbolic component produces an interpretable program.

**Neurosymbolic/differentiable program synthesis.** Existing program synthesis approaches are often trained in a supervised fashion (Gulwani, 2011; Wang et al., 2017; Shah et al., 2020), or within a (generative) policy learning context with an explicit reward function (Chen et al., 2018; Verma et al., 2018; Inala et al., 2020; Feinman & Lake, 2020). In terms of unsupervised program synthesis, the closest related area is generative modeling, as the goal there is to discover programs that can generate the training data (Ellis et al., 2018; Feinman & Lake, 2020) – which can be viewed as analogous to learning a symbolic decoder rather than a symbolic encoder. Studying how to incorporate such methods into our framework can be an interesting future direction.

**Representation learning for behavior analysis.** Representation learning has been applied to a variety of downstream tasks for behavior analysis, such as discovering behavior motifs (Berman et al., 2014; Singh et al., 2021), identifying internal states (Calhoun et al., 2019), and improving sample-efficiency (Sun et al., 2021b). Studies in this area have used methods such as VAE (Kingma & Welling, 2014), AR-HMM (Wiltschko et al., 2015), and Umap (McInnes et al., 2018) to better understand the latent structure of behavior. Similar to a few other representation learning methods (Luxem et al., 2020; Sun et al., 2021b), we also use an encoder-decoder setup on trajectory data. However, our work learns a neurosymbolic encoder whereas existing works in this area have fully neural encoders. Our work can aid behavior analysis by learning more interpretable latent representations and can be applied to existing frameworks, such as task programming.

## 6 CONCLUSION

We present a novel approach for unsupervised learning of neurosymbolic encoders. Our approach integrates the VAE framework with program synthesis and results in a learned representation with both neural and symbolic components. Experiments on trajectory data from behavior analysis demonstrate that our programmatic descriptions of the latent space result in more meaningful clusters relative to human-defined behaviors, compared to purely neural encoders. Additionally, we show the practicality of our approach by applying our learned programs to achieve comparable performance to expert-constructed tasks in a self-supervised learning approach for behavior classification.

Based on our work, there are many future directions to explore for neurosymbolic encoders. One direction is based on the observation that interpreting multiple programs simultaneously can still be difficult for domain experts. Combining our approach with other clustering methods, such as hierarchical clustering (Rokach & Maimon, 2005), or working with domain experts to detail more expressive DSLs can help with interpretability. Another direction is to extend this work to other domains such as image and text data, to study both discrete and continuous symbolic latent representations in a more naturally interpretable way. A third direction is to improve upon our greedy approach in Algorithm 2 for finding the optimal set of symbolic programs, e.g. by performing local coordinate ascent in program space, similar to algorithms for large-scale neighborhood search (Ahuja et al., 2002). Lastly, while practically-oriented extensions of VAEs such as our own have yielded great practical benefit, they often lead to sub-optimal results from a pure likelihood (or ELBO) perspective. One final direction is to rigorously formulate a learning objective from the ground up that formally encapsulates practically-oriented extensions of VAEs.

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
