# OpenReview forum: "Unsupervised Learning of Neurosymbolic Encoders"
_ICLR.cc/2022/Conference — ICLR 2022 Submitted_

### Official Review · Reviewer_G4ix · 2021-10-31

**Correctness:** 3
**Technical Novelty And Significance:** 3
**Empirical Novelty And Significance:** 2
**Recommendation:** 5
**Confidence:** 3

**Main Review:**

The strength of this paper is described above. Although the concept and method are simple and new, and the paper itself is well-written, the significance in the problem setting and experimental results were unclear to me at this stage. The specific comments are as follows.

Major concerns:
1. Although the unsupervised neurosymbolic “encoder” is new (as described in Sec. 5, but the decoders existed in [Feinman & Lake, 2020] and also the following [1]?), but I did not understand why the authors considered that creating the encoder is important.
2. Although the concept of the proposed method is general and interesting, the applied domains were limited to animal/human trajectories. That is, it is unclear to me why the authors selected these domains to verify the proposed method. Or verification using other domain data may be also required.
3. The significance of the experimental results was unclear to me. The task to extract distinctive information between the data w/ and w/o interaction in mice may be considered less challenging for me. The basketball task including offensive/defensive players is more challenging, but the result of the proposed method was overall comparable with the baselines (whereas it improved slightly with respect to purity). In the basketball task, I am also concerned about the importance of offensive/defensive players detection (this is usually given). The final downstream task (not tasks) is a classification task, and the results were comparable with the previous work (Sun et al. 2021b).

Minor comments:
4. What is x in Sec. 2.2? If it is some inputs, what types of inputs?
5. Which is correct, .428 in NMI of Figure 4 or .423 of Table 1? Or What DSL was used in Table 1 results?

[1] Halley Young, Osbert Bastani, Mayur Naik, Learning Neurosymbolic Generative Models via Program Synthesis, ICML, 97:7144-7153, 2019.

**Summary Of The Paper:**

The authors propose an unsupervised approach to train neurosymbolic encoders to obtain a programmatically interpretable representation using the dictionary of a domain-speciﬁc language (DSL). The experimental results show that the proposed method can outperform baseline neural encoders in extracting semantically meaningful representations of behavior in CalMS21 (mice) dataset. The results also show that the performance can be robust across different DSL designs by domain experts.






**Summary Of The Review:**

Although the concept and method are simple and new, and the paper itself is well-written, the significance in the problem setting and experimental results were unclear for me. Therefore, it is difficult for me to provide a higher rating at this stage.

---

> ### Author Response · Authors · 2021-11-23
> **Response to Reviewer G4ix**
>
> Thank you very much for your feedback! We address your comments below:
>
> > “... did not understand why the authors considered that creating the encoder is important.”
>
> Learning semantically interpretable encoders is important for two reasons.  Firstly, unsupervised clustering is arguably the most important data analysis bottleneck in behavioral neuroscience, and recent work such as in [1] have highlighted the challenge in interpreting learned clusters. Secondly, programmatic encoders are effectively self-supervised labeling functions. Many recent works have shown how to leverage expert-written programs as a form of weak supervision [2], and we can immediately use our learned encoders as a drop-in replacement.  In other words, we can automatically learn self-supervision tasks instead of hand-crafting them (see Figure 5). For these reasons, we view learning neurosymbolic encoders as important.
>
> > “... it is unclear to me why the authors selected these domains to verify the proposed method.”
>
> We focus on trajectory data because behavior modeling is an application area that already aims to leverage domain knowledge expressed as programs. This allows us to integrate our learned programs into existing behavior analysis frameworks such as task programming [2] in Figure 5). Generally, we are very interested in extending our approach to other domains, which is a direction that we are actively exploring. For example, we first need to define a way to efficiently represent visual domain expert knowledge in an interpretable way to encode symbolic knowledge on natural images. We view this line of work as complementary to our current approach.
>
> > “The significance of the experimental results was unclear to me.”
>
> The ultimate test of significance is the usefulness of our learned programs. Our first evaluation method is to use cluster metrics, which we acknowledge is just a proxy for our true goal and doesn’t capture the entire picture. Nevertheless, for this evaluation method we view that merely learning unsupervised and interpretable clusters that match up well with human-annotated ones is already a success.
>
> The Task Programming framework of (Sun et al., 2021b) is a self-supervised learning approach that then transfers the learned representation to a classification task.  Task programming uses hand-crafted programs for self-supervision, whereas Figure 5 shows a comparison where we alternatively used our automatically learned programs.  The fact that the results are comparable is thus significant because we essentially automatically learned the self-supervision tasks.
>
> > “Which is correct, .428 in NMI of Figure 4 or .423 of Table 1?”
>
> 0.423 NMI in Table 1 is the median results from three runs. 0.428 is the NMI of the particular program shown in Figure 4 (a single run).
>
> [1] Luxem et al., Identifying behavioral structure from deep variational embeddings of animal motion.
>
> [2] Sun et al., Task Programming: Learning Data Efficient Behavior Representations, CVPR 2021.

---

> > ### Comment · Reviewer_G4ix · 2021-11-26
> > **Thanks for the responses**
> >
> > Thanks for the responses.
> > I understood the importance of learning neurosymbolic encoders.
> > For behavior modeling from trajectories, I agree with the necessity of leveraging domain knowledge in the domain. However, at this stage, it was not carefully described in the introduction and the title is too general. Of course, the application to the image will enhance this paper.
> > Regarding the experiment results, I understood the significance of comparing to the task programming framework of (Sun et al., 2021b). However, my concerns were about other tasks themselves and comparisons with TVAE variants in Table 3 and they are still unclear.
> > Finally, the difference between \mathrm{x} in Sec 2.1 and x in Sec 2.2 was still unclear.
> > Although the concept and method are interesting, in my opinion, the presentation can be improved for obtaining a higher rating.

---

> > > ### Author Response · Authors · 2021-11-29
> > > **Thanks for your update!**
> > >
> > > We thank the reviewer for your response, and we provide additional clarifications below.
> > >
> > > > However, my concerns were about other tasks themselves and comparisons with TVAE variants in Table 3 and they are still unclear.
> > >
> > > Similar to other papers on unsupervised learning ([1, 2, 3]), we evaluate our learned programs and clusters against human-defined classes. This is a common way to evaluate unsupervised learning since human-annotated classes provide a proxy on what humans consider to be reasonable clusters in the data. We want to train our model to discover the similar structure from data.
> > >
> > > Previous TVAE variants are not able to leverage domain knowledge (such as those used in behavioral neuroscience (Segalin et al., 2020)), and thus produce clusters that do not correspond well to pre-defined behavioral classes from domain experts (Table 1). The symbolic approach is crucial in this case to leverage the structured domain knowledge in the DSL to produce clusters that better agree with human-annotated behaviors (Table 1). Furthermore, symbolic encoders are necessary to produce interpretable programs of the clusters (previous TVAE variants are not able to do this) - the programs produced by our method can be used by human experts studying behavior (Section 4.2.3, Question 1, second paragraph).
> > >
> > > > Finally, the difference between \mathrm{x} in Sec 2.1 and x in Sec 2.2 was still unclear.
> > >
> > > We use \mathbf{x} to refer to input data, and it should be consistent for the entire paper. Thank you for pointing out this typo.
> > >
> > >
> > > [1] Luxem et al., Identifying behavioral structure from deep variational embeddings of animal motion.
> > >
> > > [2] Caron et al., Deep Clustering for Unsupervised Learning of Visual Features, ECCV 2018
> > >
> > > [3] Yang et al., Joint Unsupervised Learning of Deep Representations and Image Clusters, CVPR 2016

---

### Official Review · Reviewer_a6JR · 2021-11-03

**Correctness:** 3
**Technical Novelty And Significance:** 3
**Empirical Novelty And Significance:** 3
**Recommendation:** 5
**Confidence:** 3

**Main Review:**

I think the idea of utilizing self-supervised training task to generate symbolic programs is very interesting and worth exploring.

My major concerns lie in the approach and evaluation.

For approach:
1) Though I read the paper carefully, I still didn't fully get how the authors represent the program via z(alpha, fi). It seems that the output of VAE should be a vector, and how did the authors decode z(alpha, fi) to alpha and fi, especially when alpha should be a "discrete program architecture". Is alpha a program or just a single operator? Also how did you encode q(alpha, fi) when alpha could be a complex tree-structured program?
2) I'm also not fully understand equation 3. How to you search the child program of alpha in a differentiable manner? Why this is a supervised task (from my understanding it's more like finding a child program that outputs similar encoding vector as parent program).
3) You mention you "repeat steps 1&2 until q(apha, fi) is fully symbolic". The meaning of fully symbolic is really confusing to me. What is the stopping criterion? Also, what's the intuition that you need to update the parameter agrain after fixing program alpha?
4) The authors didn't mention how each function in the DSL is implemented in paper and appendix. I think it's very important to understand the whole architecture. Also, it's important to know how you handle the non-differentiable operators, such as if-then, and others.


For evaluation:
1) Current evaluation seems not very convincing to me. The authors only show that with the help of symbolic program, the method could get representations with better cluster quality (program helps representation learning). But I think a more intersting perspective is to see whether the learned program itself is helpful. For example, whether it could be used to predict future trajectory (such as 3-body problem), or even help solving some high-level reasoning tasks.
2) Lack of the ablation study of the proposed framework. For example, why it's important to update the parameter again. Is the design of DSL influences the final representation learning, etc.

**Summary Of The Paper:**

This paper tries to use VAE-based generative task to generate a latent representation that could be encoded as symbolic program. The authors show that in this way it could achieve better cluster results and also generate interpret-able and reasonable programs for each data.

**Summary Of The Review:**

This paper proposes a very interesting research direction, but the writing and organization of the proposed method make it hard to understand it. In addition, the current evaluation is way too simple and not interesting enough. I highly recommend the authors to add some experiments to show that the learned programs can help some down-stream tasks.

---

> ### Author Response · Authors · 2021-11-23
> **Response to Reviewer a6JR**
>
> Thank you very much for your feedback! We address your comments below:
>
> > “... how the authors represent the program via z(alpha, fi) ... Is alpha a program or just a single operator?”
>
> Programs in our work are represented as a tuple (alpha, fi). Alpha refers to the program architecture (optimized via search) and fi are learnable parameters (optimized via stochastic gradient descent with alpha fixed). By analogy, in neural architecture search, alpha would refer to the network architecture, and fi the continuous parameters.  The execution of the program produces a vector-valued mean of the latent variable z.  Following standard VAE notation, z_(alpha,pi) means that (alpha,pi) are the parameters of the function that defines the distribution of z.
>
> > “I'm also not fully understand equation 3.”
>
> Your understanding is correct. We perform this search by enumerating and searching through a program’s children (in our experimental setup, programs have at most 8 children). We acknowledge that more sophisticated search procedures may be required as the search space of programs grows, but we leave this as a direction for future work.  By analogy to neural architecture search, one could also search for the neural network architecture when training a neural encoder.  If one solves the VAE optimization problem in alternating fashion then one can freeze the decoder and search for the architecture of the encoder via a direct training signal (which looks like supervised learning from the encoder’s perspective).
>
> > “... The meaning of fully symbolic is really confusing to me. What is the stopping criterion? Also, what's the intuition that you need to update the parameter ...?”
>
> By fully symbolic, we mean when a program has no more nonterminals - i.e. no more child programs. The intuition for updating the parameters is that the assignment of discrete latent clusters can shift given a new, more structured program. We found that this iterative procedure led to more similar learned programs across different runs.
>
> > “... how each function in the DSL is implemented ... how you handle the non-differentiable operators.”
>
> We agree that this is important and will add this to the appendix. For if-then-else, we use a smooth approximation, as done in [1]
>
> > “Current evaluation seems not very convincing to me.”
>
> We demonstrate usefulness in Figure 5, which shows the performance of our learned programs when integrated into the Task Programming framework [2].  Task programming is a state-of-the-art behavior representation learning approach that uses expert-written programs to define self-supervision tasks. We instead use our automatically learned programs, and demonstrate comparable performance, which we view as an advancement on the state-of-the-art.
>
> > “Lack of the ablation study of the proposed framework.”
>
> We do perform several ablation studies in section 4.2.3, such as the sensitivity to DSL and using the DSL features as part of the input instead. We provide intuition for why it’s important to update the parameters, but can include that experiment as well.
>
> [1] Shah et al., Learning Differentiable Programs with Admissible Neural Heuristics, NeurIPS 2020.
>
> [2] Sun et al., Task Programming: Learning Data Efficient Behavior Representations, CVPR 2021.

---

### Official Review · Reviewer_zt4F · 2021-11-03

**Correctness:** 4
**Technical Novelty And Significance:** 2
**Empirical Novelty And Significance:** Not applicable
**Recommendation:** 6
**Confidence:** 4

**Main Review:**

The paper is clearly written and the idea was easy to understand and
follow. The experiments are always going to be a challenge in a
space like this as how can we say if the symbolic latent
representation meaningfully captures something in the data.
The synthetic result while a little contrived is convincing that
at least in a controlled setting a reasonable program is produced.
I wish there was more exploration of how often are sensible
programs learned and how sensitive this is to the choice of DSLs.
Maybe something comparing the programs of experts to what
the latent representation learned? Basically the program equivlaent
of showing a grid of faces.

This would also benefit from comparisons that use purely the neural component

I am also curious empirically how it's decided how many symbolic
programs should be in the latent representation? Would a similar
process be used for real-world data?

My main concern is how heavily this paper relies on the NEAR
work particularly for learning the symbolic encoder and even
the experimental data used. As the VAE bits are fairly standard
it makes the work feel fairly incremental.


**Summary Of The Paper:**

This paper is a clean extension of the supervised neurosymbolic approach
of Shah et al. to the unsupervised setting where a VAE is used
to encode input to the system as an interpretable program.
These latent representations are shown to be interpretable for
a synthetic experiment and useful for downstream tasks.


**Summary Of The Review:**

This is an interesting and novel way to learn programs as symbolic encoders of the input. I have some nagging concerns about how incremental the contribution given how much this work relies on Shah et al.

---

> ### Author Response · Authors · 2021-11-23
> **Response to Reviewer zt4F**
>
> Thank you very much for your feedback! We address your comments below:
>
> > "... how often are sensible programs learned and how sensitive this is to the choice of DSLs.”
>
> For studying how sensible the programs are, we do this quantitatively by comparing the learned program clusters against baselines (Table 1) as well as qualitatively by studying domain expert interpretation of the programs (Section 4.2.3, paragraph 2).
>
> For studying sensitivity to the DSLs, we do this by using DSLs produced by different domain experts (Table 2), and find that using any of the provided DSLs, our method outperforms baselines.
>
> > “This would also benefit from comparisons that use purely the neural component”
>
> This comparison is provided in our paper vs. JointVAEs (which uses neural components for learning both discrete and continuous representations), see Table 1. We will update the text so this is more clear.
>
> > “... how it's decided how many symbolic programs should be in the latent representation?”
>
> The number of symbolic programs would depend on the domain and the desired number of clusters to be studied in the domain (this is similar to other clustering algorithms with a hyperparameter for the number of clusters). Since our algorithm produces programs iteratively (Algorithm 2), users can start by producing one symbolic program, and iteratively produce more programs to get a more fine-grained clustering of the data.
>
> We would also like to note that we have already applied our framework to real-world data from behavioral neuroscience and sports analytics (Section 4.2), and we study the effect of varying the number of programs (Table 1).
>
> > “... how heavily this paper relies on the NEAR”
>
> Our approach relies on NEAR much like how deep learning relies on Adam.  In other words, we view NEAR as an optimization engine, and our overall approach (described in Figure 1) transcends NEAR.  We certainly appreciate that any work can seem incremental if viewed solely as putting together pieces of prior work.  Indeed, VAEs themselves might seem incremental because they put together variational inference with the reparameterization trick.  To the best of our knowledge, our approach is the first ever to learn neurosymbolic encoders, which we view as novel.

---

### Official Review · Reviewer_cxLd · 2021-11-14

**Correctness:** 4
**Technical Novelty And Significance:** 3
**Empirical Novelty And Significance:** 2
**Recommendation:** 5
**Confidence:** 3

**Main Review:**

Positives
+ The general problem of learning programs to represent data in unsupervised learning is novel and potentially fundamental to machine learning and AI
+ The proposed method is simple and interesting, and seems to achieve nice gains over purely neural approaches
+ The paper is generally well written

Negatives
- My main issue is that the neuroscience data that the paper currently mainly evaluates on is not something a general ML audience might be very familiar with, and thus, it is hard to estimate what makes the task hard or easy. Could the authors give intuitions on why regular neural encodings fail to cluster in a better manner? Is it because of low SNR in the sequence data, or that RNN encoders pick up on spurious noise signals and thus do not yield a sufficiently discriminative embedding? If that is the case how would a convolutional encoder perform? Are the gains from the symbolic approach something that could just be achieved by a convolutional encoder? How much do we really need the symbolic approach and what are the inductive biases that we gain from them? I think these are important questions to answer in order to judge the efficacy of the proposed approach. (*)

- In terms of the methodology, I do not understand why increasing the complexity of the programs gradually also leads to the purely neural part becoming less and less informative in the toy-dataset. Could it not have been the case that somehow the more complex program just learnt the features that the less complex program (with say length 2) learnt, and that in going from Fig. 3 d) to 3 e) we just learnt the same embedding space for the neural part? Is it encouraged that we learn more complex programs when we increase the length of the programs by tuning the corresponding gamma weights in Eqn. 5, or the corresponding channel capacities? (*)

- I generally think such an encoding is potentially very useful for image data, and might be a big step towards better image representations and advances towards human-like intelligence. It would be great if the authors could comment on the potential for extending the approach to such applications and how far we might be from such use-cases.

Minor Points:
1) [A] is a closely related work that would be useful to cite in a neural-symbolic + amortized inference / VAE context.

[A]: Vedantam, Ramakrishna, Karan Desai, Stefan Lee, Marcus Rohrbach, Dhruv Batra, and Devi Parikh. 2019. “Probabilistic Neural-Symbolic Models for Interpretable Visual Question Answering.” arXiv [cs.LG]. arXiv. http://arxiv.org/abs/1902.07864.

**Summary Of The Paper:**

The paper proposes to learn a novel, interpretable neuro-symbolic encoder for sequences in an autoencoding framework. The key idea is that one learns both a symbolic as well as a neural encoder, and gradually makes the symbolic encoder more and more structured progressively increasing the complexity of it. Given such an encoder, the paper proceeds to show that the encoding from it is useful for clustering sequence data and demonstrates gains over other “unstructured” purely neural approaches.


**Summary Of The Review:**

My main concerns with the paper are around whether the RNN encodings are overfitting and picking up on spurious correlations in sequence data that say a convolutional encoder might already fix, which will mean that one does not really need symbolic encoders for the current task. This is an even larger concern in light of the fact that a general machine learning researcher might not have a lot of great intuitions about the neuroscience sequence data and that makes it hard to assess the impact of the work. For the rebuttal I would encourage the authors to address concerns marked with (*) in the main review.

---

> ### Author Response · Authors · 2021-11-23
> **Response to Reviewer cxLd**
>
> Thank you very much for your feedback! We address your comments below:
>
> > “Could the authors give intuitions on why regular neural encodings fail to cluster in a better manner?”
>
> Regular neural encodings are not able to leverage existing domain knowledge (such as those used in behavioral neuroscience (Segalin et al., 2020)), and thus produce clusters that do not correspond well to pre-defined behavioral classes from domain experts (Table 1). We expect convolutional encoders would have similar problems. The symbolic approach is crucial in this case to leverage the structured domain knowledge in the DSL to produce clusters that better agree with human-annotated behaviors (Table 1). Furthermore, symbolic encoders are necessary to produce interpretable programs of the clusters (neither RNN nor convolutional encoders are able to do this) - the programs produced by our method can be used by human experts studying behavior (Section 4.2.3, Question 1, second paragraph).
>
> More concretely, here we evaluate the neural encoding clusters based on behaviors annotated by neuroscientists (4 behaviors = attack, sniff, mount, other) (Section 4.2.1). There are behavioral features that are useful for identifying these behaviors that have already been identified by domain experts (Segalin et al., 2020) - for example, distance between mice is a behavioral feature that’s useful for identifying whether the mice are interacting, such as sniffing. Neural encodings are not able to leverage this, while our symbolic approach uses this domain knowledge through the DSL.
>
> > "... why increasing the complexity of the programs gradually also leads to the purely neural part becoming less and less informative in the toy-dataset.”
>
> The intuition between the neural and symbolic components is that the neural component captures residual information not encoded by the symbolic encoders (Section 3, paragraph 1) and the adversarial loss (Section 3.3) encourages the encoded neural & symbolic components to be factorized. The synthetic dataset is a demonstration of this, showing the neural embedding space for 0,1,2 learned symbolic programs (as the neural embedding space contains less information on final X & Y position encoded into the symbolic programs). The learned programs for the synthetic dataset are in Figure 3b) and it demonstrates that the two learned programs are using different features (one for X position and one for Y position). We are not increasing the complexity of any individual programs in this experiment, but rather learning more programs to produce more clusters in the discrete latent space (thus removing more information from the neural continuous latent space). Here we did not tune the gamma weights in Eq (5) or adjust the channel capacity, but we used Algorithm 2 to discover an additional program for the symbolic component (so 2 programs for both X and Y position, instead of just 1 program on X position).
>
> > “... the potential for extending the approach to such applications and how far we might be from such use-cases.”
>
> We appreciate this suggestion, and this is a direction we are really interested in further exploring. For image representations, we need a way to efficiently represent visual domain expert knowledge in an interpretable way and develop ways to encode symbolic knowledge on natural images. We view this line of work as complementary to our current approach.
>
> We would like to note that we chose to focus on trajectory data in this work since the current application area is already useful (for example, our method can be directly integrated into existing behavior analysis frameworks such as task programming in Figure 5) and our datasets are representative of real behavior data used by real domain scientists (CalMS21 collected by behavioral neuroscientists to study mouse social behavior; Basketball collected by sports analysts from real basketball games), and were used in real domain applications before use in machine learning research.

---

### Author Response · Authors · 2021-11-23
**Global comments for all reviewers**

We thank all reviewers for their helpful feedback. We respond to the main criticisms below, as well as emphasize the significance of our contributions.

1) Why neurosymbolic encoders?

While we are aware of previous work in learning programmatic decoders, we highlight that learning semantically interpretable encoders is important for two reasons.  Firstly, programmatic encoders can produce interpretable clusters, which helps tackle one of the most important bottlenecks in behavior discovery: interpreting behavioral clusters. Recent work such as in [1] have highlighted the challenge in interpreting learned clusters. Secondly, programmatic encoders are effectively self-supervised labeling functions. Many recent works have shown how to leverage expert-written programs as a form of weak supervision [2], and we can immediately use our learned encoders as a drop-in replacement.  In other words, we can automatically learn self-supervision tasks instead of hand-crafting them (see Figure 5). For these reasons, we view learning neurosymbolic encoders as important.

2) Significance of experimental results

The ultimate test of significance is the usefulness of our learned programs. Our first evaluation method is to use cluster metrics relative to human-defined behaviors, which we acknowledge is just a proxy for our true goal (learning meaningful programs) and doesn’t capture the entire picture. Nevertheless, for this evaluation method we view that merely learning *unsupervised* and *interpretable* clusters that match up well with *human-annotated* ones is already a success.

The Task Programming framework of [2] is a self-supervised learning approach that then transfers the learned representations to a classification task.  Task programming uses *hand-crafted* programs for self-supervision, whereas Figure 5 shows a comparison where we alternatively used our *automatically learned programs*.  The fact that the results are comparable is thus significant because we essentially automatically learned the self-supervision tasks (instead of hand-crafting them, as in Task Programming).

Lastly, reviewers asked for ablation studies, which we have already included in the main paper, such as sensitivity to choice of DSL (section 4.2.3) and comparison with a fully neural representation (JointVAE in Table 1).

3. Future impact in other domains

We agree with reviewers that extending our work to other domains is an important direction for future work. We focus on trajectory data because behavior modeling is an application area (with real-world applications, such as in behavioral neuroscience and sports analytics) that already aims to leverage domain knowledge expressed as programs, which allows us to integrate our learned programs into existing behavior analysis frameworks such as task programming in Figure 5.

Although extending to other domains requires more work (e.g. how do we efficiently represent visual domain expert knowledge in an interpretable way to encode symbolic knowledge on natural images?), we believe that our contributions towards learning programmatic and interpretable representations are already significant, and that it would be of great interest to the general ML community.


[1] Luxem et al., Identifying behavioral structure from deep variational embeddings of animal motion.

[2] Sun et al., Task Programming: Learning Data Efficient Behavior Representations, CVPR 2021.

---

### Decision · Program_Chairs · 2022-01-20

**Decision:**

Reject

**Comment:**

The authors develop a technique for unsupervised learning of neurosymbolic encoders. Some of the difficulty with the paper came from the accessibility to a broader machine learning audience, though there is related work such as Shah 2020 in machine learning. The other difficulty came from the experiments: there was both a question about the metrics and the task. Quoting a reviewer

"Current evaluation seems not very convincing to me. The authors only show that with the help of symbolic program, the method could get representations with better cluster quality (program helps representation learning). But I think a more intersting perspective is to see whether the learned program itself is helpful. For example, whether it could be used to predict future trajectory (such as 3-body problem), or even help solving some high-level reasoning tasks."

and another reviewer

"Maybe something comparing the programs of experts to what the latent representation learned?"

Making the paper more accessible and improving the experiments will improve its quality.